# Comparative Evaluation of Mineralized Bone Allografts for Spinal Fusion Surgery

**DOI:** 10.3390/jfb14070384

**Published:** 2023-07-21

**Authors:** Paul J. Hubbell, Brandon Roth, Jon E. Block

**Affiliations:** 1Southern Pain and Neurologic, 3939 Houma Blvd., Building 2, Suite 6, Metairie, LA 70006, USA; 2AZ Pain Doctors, 14420 W Meeker Blvd., Building A, Ste. 211, Sun City West, AZ 85375, USA; 3Independent Consultant, 2210 Jackson Street, Suite 401, San Francisco, CA 94115, USA

**Keywords:** bone, allograft, homologous, spine, structural

## Abstract

The primary objective of this review is to evaluate whether the degree of processing and the clinical utility of commercially available mineralized bone allografts for spine surgery meet the 2020 US Food and Drug Administration’s (FDA) guideline definitions for *minimal manipulation* and *homologous use*, respectively. We also assessed the consistency of performance of these products by examining the comparative postoperative radiographic fusion rates following spine surgery. Based on the FDA’s criteria for determining whether a structural allograft averts regulatory oversight and classification as a drug/device/biologic, mineralized bone allografts were judged to meet the Agency’s definitional descriptions for *minimal manipulation* and *homologous use* when complying with the American Association of Tissue Banks’ (AATB) accredited guidelines for bone allograft harvesting, processing, storing and transplanting. Thus, these products do not require FDA medical device clearance. Radiographic fusion rates achieved with mineralized bone allografts were uniformly high (>85%) across three published systematic reviews. Little variation was found in the fusion rates irrespective of anatomical location, allograft geometry, dimensions or indication, and in most cases, the rates were similar to those for autologous bone alone. Continued utilization of mineralized bone allografts should be encouraged across all spine surgery applications where supplemental grafts and/or segmental stability are required to support mechanically solid arthrodeses.

## 1. Introduction

Allogeneic bone, commonly referred to as a bone allograft, is human cadaveric donor tissue that is transplanted to a recipient host as part of an arthrodetic surgical procedure. Due to the volume limitations and morbidity associated with autologous bone graft harvesting [1,2], bone allografts have become indispensable in spine surgery to support and facilitate fusion, and maintain mechanical intersegmental stability [3].

Bone allografts exhibit various morphological characteristics and biochemical properties due, in large part, to the degree of post-procurement tissue processing [4]. Osseous tissue properties such as the surface topography and structure, biochemical composition and mechanical stability may be altered by the processing of the graft [5]. In fact, the amount and type of manipulation of donor skeletal tissue dictates the commensurate regulatory pathway in the US for bone allografts; it ranges from minimally manipulated bone regulated as a “tissue” solely under section 361 of the Public Health Service (PHS) Act, and regulations in 21 Code of Federal Regulations (CFR) Part 1271, to more highly-manipulated bone regulated as a drug, device and/or biological product regulated under the Federal Food, Drug and Cosmetic (FD&C) Act and/or section 351 of the PHS Act [6]. These distinct regulatory pathways have profoundly different ramifications with respect to the comparable product mechanisms of action, performance characteristics, utility, commercial claims and the reimbursement status.

This manuscript provides the rationale for defining minimally manipulated, mineralized structural bone allografts for homologous use and differentiating these products from other derivate allogeneic bone products that undergo more extensive tissue processing.

## 2. Methods

We provide a historical perspective for bone allograft usage, as well as synthesize the radiographic and clinical findings across studies evaluating allograft safety and efficacy for spinal surgery applications.

## 3. Bone Allograft Characteristics, Processing, Types and Indications

Characterizing a bone allograft product as a tissue or a drug/device/biologic is based on a tiered, risk-based approach, and hinges on several essential criteria, as specified by the United States Food and Drug Administration (FDA) in their 2020 guidance document entitled *Regulatory Considerations for Human Cells, Tissues, and Cellular and Tissue-Based Products: Minimal Manipulation and Homologous Use* [6]. Bone allograft products avert classification and regulatory oversight as a drug/device/biologic if their tissue processing involves *minimal manipulation* [21 CFR 1271.3(f)] and their use is *homologous*, as defined as the *repair, reconstruction, replacement, or supplementation of a recipient’s cells or tissues with allogeneic human cells, tissues, and cellular and tissue-based products (HCT/Ps) that perform the same basic function or functions in the recipient as in the donor* [21 CFR 1271.3(c)].

Following strict donor screening protocols for procurement, mineralized bone allografts must undergo, as an initial step, a thorough removal and/or attenuation of potentially antigenic material, such as bone marrow, collagen or potential pathogens to assure safe and effective transplantation [7]. This mandatory tissue cleansing process involves a rigorous protocol of physical, chemical, and sterilization procedures but does not alter the inorganic composition or crystalline structure of the native bone tissue [8]. 

Physical cleaning and decellularizing processes include temperature manipulation, sonication, centrifugation, high-pressure water jet washing, and hydrostatic and atmospheric pressure adjustments. Chemical-based cleaning processes include the use of alcohol, hydrogen peroxide, detergents, and surfactants. Following bone cleaning, a sterilization protocol is required to further assure deactivation of pathogens. Bone allografts are stored frozen, freeze-dried or processed directly into smaller pieces under aseptic or clean-room conditions.

Classification of these tissue processing procedures for mineralized bone allografts corresponds with FDA’s definition of *minimal manipulation* of structural tissues. Specifically, minimal manipulation involves *processing that does not alter the original relevant characteristics of the tissue relating to the tissue’s utility for reconstruction, repair, or replacement* [21 CFR 1271.3(f)] [6]. The relevant FDA guidance with reference to minimal manipulation for mineralized bone allografts is as follows:
*“Example 10-1: Original relevant characteristics of bone relating to its utility to support the body and protect internal structures include strength, and resistance to compression. Milling, grinding, and other methods for shaping and sizing bone may generally be considered minimal manipulation when they do not alter bone’s original relevant characteristics relating to its utility to support the body and protect internal structures.*
*a* *A manufacturer performs threading and other mechanical machining procedures to shape bone into dowels, screws, and pins. The HCT/Ps are generally considered minimally manipulated because the processing does not alter the bone’s original relevant characteristics relating to its utility to support the body and protect internal structures.**b* *A manufacturer grinds bone to form bone chips and particles. The HCT/Ps would generally be considered minimally manipulated because the processing does not alter the bone’s original relevant characteristics relating to its utility to support bodily structures (Section III, B, 5.).”*

In contrast, the processing required to manufacture allograft bone derivates such demineralized bone matrix (DBM), and the more recently introduced cellular bone matrices (CBM) that incorporate mesenchymal stem cells generally do not meet the FDA criteria for minimal manipulation. These materials must not be combined with other articles except water, crystalloids or a sterilizing, preserving or storage agent. The relevant FDA guidance with reference to minimal manipulation for DBM and CBM is as follows:
*c* *“A manufacturer exposes bone to acid at elevated temperature to demineralize bone and dissolve collagen in order to form a gel. The HCT/P is generally considered more than minimally manipulated because the processing alters the bone’s original relevant characteristics relating to its utility to support the body and protect internal structures (Section III, B, 5.).”*

Classification of mineralized bone allografts that have undergone minimal manipulation also correspond with the FDA’s definition of *homologous use* for structural tissues. Specifically, homologous use is defined as the *repair, reconstruction, replacement, or supplementation of a recipient’s cells or tissues with an HCT/P that performs the same basic function or functions in the recipient as in the donor* [21 CFR 1271.3(c)] [6]. The relevant FDA guidance with reference to homologous use for mineralized bone allografts is as follows:
*“Example 20-2: The basic functions of bone are supporting the body and protecting internal structures such as the brain. Allogeneic mineralized or demineralized cortical human bone is used to supplement the recipient’s bone for repair, replacement, and reconstruction of bony voids or gaps involving the extremities, cranium, and spinal column; or for augmentation for posterior lateral fusions in the spinal column. These are homologous uses because in all locations, the HCT/P is supplementing the recipient’s bone, for the purpose of supporting the body or protecting internal structures (Section IV, 3, 20.).”*

Although mineralized bone allografts, as described above, meet the FDA’s definitions of minimal manipulation and homologous use, and are, thus, not subject to regulatory oversight, there is a large body of basic science and clinical evidence to support the safe and effective use of these products in spine surgery [3,9,10,11]. That said, the limitations of these products should also be recognized. The cleansing and decellularization processes use detergents and chemicals, such as hydrogen peroxide, that act to inhibit key proteins that induce bone formation, such as bone morphogenetic proteins; these impact the performance of allografts after implantation. Freeze drying and terminal sterilization also attenuate the bone-forming ability of the allograft. In the end, the cleansing processes, particularly those using chemical-based agents as noted above, necessary to render the allograft non-antigenic, result in vital growth factors being removed; this leads to little to no osteoinductivity. The mechanical strength of bone allografts can also be reduced by freeze drying, as an example, lowering its capacity to absorb energy, increasing its brittleness, and slowing the bone integration process compared to that of bone autografts [12,13]. Minimally manipulated mineralized bone allografts are osteoconductive, biologically inert grafts [14], with slow incorporation [15]. Consequently, supplemental fixation with metallic instrumentation is routinely utilized in conjunction with bone allografts to provide immediate stabilization and graft site reinforcement to support the attenuated osseointegration process.

These grafts can be machined and fabricated into rings, wedges, struts, blocks, cubes, dowels, chips and spacers, and can be ground and morselized. Bone grafts can be primarily cortical, cortico-cancellous or cancellous in composition, each exhibiting lesser degrees of structural support, respectively [4,16]. Due to their recognized importance in spinal surgery, commercially available bone allografts products account for >50% of the non-autologous bone utilization [12]. Figure 1 shows an array of bone allograft products typically used for spine surgery applications.

Given the plethora of available configurations, mineralized bone allografts are widely indicated for spinal surgery applications wherever there is a shortage of autologous bone graft, such as lumbar postero-lateral fusion and long-segment deformity surgery. Bone allografts also serve as alternatives to metallic instrumentation (e.g., cages), as in the case of interbody fusion, where they provide intersegmental stability while osseous incorporation occurs. Similarly, structural bone allografts can provide stability across the sacroiliac joint (SIJ) space when implanted inferiorly in a surgically created, decorticated intra-articular channel to support fusion [17].

## 4. Historical Perspective

There is a deep history of clinical use of structural bone allografts in orthopedics, dating back to the late 19th century [18,19]. As clinical demand and usage increased, the need for dedicated allograft repositories arose; by the 1940s, bone banks were established. With the founding of the American Association of Tissue Banks (AATB) in 1976, stringent donor screening, tissue procurement and processing protocols were refined and validated. The AATB accredited guidelines for bone allograft harvesting, processing, storing and transplanting are complied with by an affiliated association of more than 120 tissue banks in the US [4]. While human tissue carries an inherent yet minimal risk of disease transmission, a 2005 survey sponsored by the AATB estimated an overall allograft-associated infection rate of 0.014% [20]. Importantly, this survey preceded widespread implementation of more advanced FDA-mandated methods aimed at reducing the risk of disease transmission, including sensitive nucleic acid virus testing, as well as routine terminal sterilization.

In spine surgeries, the use of bone allografts to support arthrodeses was first envisioned and later championed by the pioneering spine surgeon Ralph Bingham Cloward, who is justly credited as the inventor of the posterior lumbar interbody fusion (PLIF) technique [21]. Cloward’s seminal 1952 article entitled *The treatment of ruptured lumbar intervertebral disc by vertebral body fusion. III. Method of use of banked bone*, revolutionized spine surgery by demonstrating similar fusion rates between patients treated with structural allografts and autografts [22]. Thus, Cloward concluded that without the need to harvest autologous bone from the iliac crest, donor site morbidity could be eliminated entirely in these patients.

## 5. Radiographic and Clinical Evidence

There have been three systematic literature reviews that evaluated studies of the use of mineralized bone allografts in spinal arthrodesis surgeries using implants of different sizes and shapes [3,10,23]. Kadam et al. [3] reviewed 39 bone allograft studies, including 14 for cervical fusions, 12 for lumbar fusions, 7 for deformities and 6 for trauma or tumors. A consistently high fusion rate was reported across all studies (median: 87%).

Tuchman et al. [23] reviewed 13 studies of cervical fusions that compared bone allografts with autologous bone. The radiographic fusion rates for patients treated with bone allografts ranged from 74% to 100% (median: 93%); the good to excellent clinical outcomes (e.g., Odom’s criteria) ranged from 73% to 93% (median: 91%). All of the allograft studies avoided donor site morbidity.

Tavares et al. [10], using meta-analytical statistical methods specified by PRISMA, evaluated surgical outcomes in 516 patients (eight studies) who underwent cervical and lumbar spinal fusion procedures with bone allografts. The pooled proportional rate demonstrated 381 fusions out of 494 (87.8%, CI 80.8–93.4%). Pseudarthrosis was rare with allograft use, with a pooled proportion of four events among 243 patients (4.8%, CI 0.1–15.7%). Adverse events such as pain, infection and graft-related events occurred infrequently among the allograft-treated patients (73 events among 459 patients; 16%), and were significantly lower than adverse event rates with autografts due to inherent donor site morbidity (860 out of 2001; 43%). Figure 2 illustrates the comparative average fusion rates, ranging from 50% to 100%, across all studies [24,25,26,27,28,29,30,31].

Additionally, McGuire et al. [11] reported a fusion rate of 89.5% (34 of 38 patients) using fibular dowel allografts implanted in the intra-articular portion of the SIJ. Successful osseointegration and solid arthrodesis across the SIJ is exhibited in Figure 3. These findings were noted as supporting rationale for the 2015 North American Spine Society’s reimbursement coverage recommendations for percutaneous SIJ fusions [32].

## 6. Interpretation

Mineralized structural bone allografts continue to play a vital role in contemporary spine surgeries. These grafts meet the FDA’s recently-published guidance definitions for minimal manipulation and homologous use and, as such, are not subject to the same regulatory oversight as bone allograft derivates such as DBM and CBM, which require more complex processing that may alter the osseous characteristics of the native tissue. Indeed, there are now over 400 510(k) medical device clearances for bone grafting products that involve more than minimal manipulation by the FDA’s criteria [4].

The radiographic fusion rates achieved with mineralized bone allograft were uniformly high (>85%) and consistent, with little variation, irrespective of anatomical location, implant geometry, dimensions or indication; in most cases, the rates achieved were similar to those with autologous bone. It should be recognized, however, that bone allografts are almost never used alone in fusion applications; they are often combined with local autologous bone, bone marrow aspirate, or DBM and supplemental fixation devices to improve the chance of osseointegration. However, the consistent clinical performance of mineralized structural bone allografts over several decades of study is in contrast to the results achieved with other commercial allograft products, such as DBM or CBM, where variability may be associated in large part with differing processing methods [33,34]. For example, Russell et al. [35] found large variations in fusion rates (0–100%) for seven commercially-available DBM products using a rat postero-lateral spinal fusion model, with two products yielding a 0% fusion rate by manual palpation and lack of evidence of bone bridging by micro-computed tomography. Similarly, Johnstone et al. [36], using the same postero-lateral spinal fusion model, reported fusion rates of 7%, 71% and 77% for three commercially-available CBM allograft products. These findings suggest that more highly manipulated bone allografts exhibit greater variability in clinical performance than minimally manipulated mineralized structural bone allografts [37].

While many different shapes and sizes of structural allografts exist as HCT/Ps under section 361 of the PHS Act, a clinical study has not been conducted to evaluate the comparative geometries of allograft implants on fusion rates for any specific clinical application. This is largely due to the fact that regardless of allograft geometry, all of these implants are minimally manipulated with inherently similar compositions, and must be used for the same purposes that exist in the donors. Continued utilization of mineralized bone allografts from AATB-accredited bone banks should be encouraged across all spinal arthrodesis applications where supplemental grafts and/or segmental stability are required, in order to support mechanically solid fusions.

## Figures and Tables

**Figure 1 jfb-14-00384-f001:**
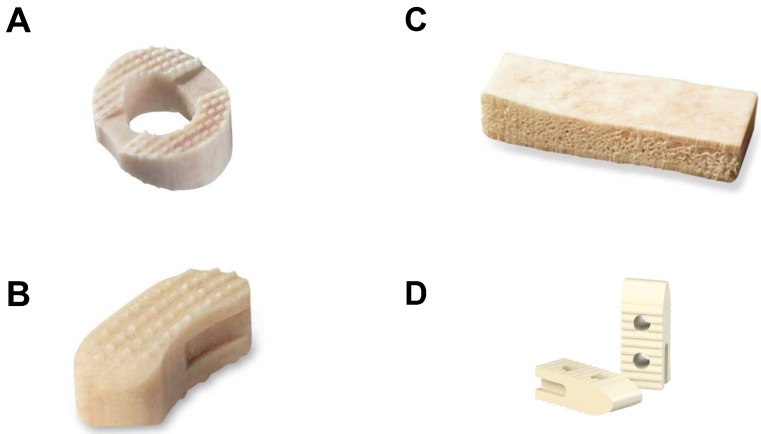
Examples of commercially available mineralized bone allograft products for cervical interbody fusion (**A**), lumbar interbody fusion (**B**), lumbar postero-lateral fusion (**C**) and intra-articular sacroiliac joint fusion (**D**).

**Figure 2 jfb-14-00384-f002:**
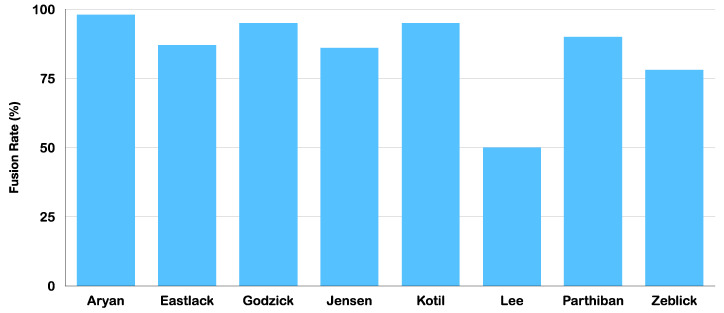
Bar graph illustrating average fusion rates across 8 studies of bone allograft usage in spine surgery as adapted from Tavares et al. [10]. The pooled proportional rate based on all patients over all studies was 87.8% (CI 80.8–93.4%).

**Figure 3 jfb-14-00384-f003:**
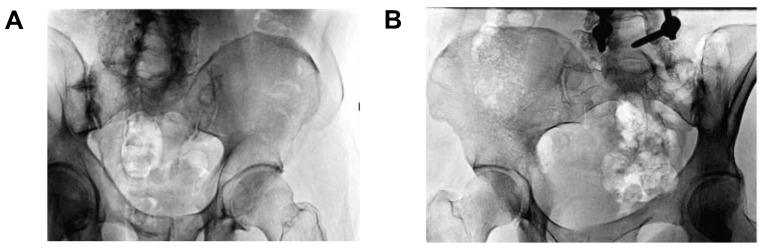
Oblique pelvic radiographs showing early osseointegration of intra-articular dowel allograft implantation at 1 year (**A**), and solid bony arthrodesis across the SIJ at 6 years postoperative follow-up (**B**).

## Data Availability

Not applicable.

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
