# Peer review of "Comparative Evaluation of Mineralized Bone Allografts for Spinal Fusion Surgery"

_jfb, 2023, doi:10.3390/jfb14070384_

Round 1

Reviewer 1 Report

The manuscript titled “Comparative evaluation of mineralized bone allografts for spinal surgery”, opines on the current status of the various mineralized bone allografts available for clinical use. The manuscript is easy to follow and would serve as a brief reference for understanding the FDA guidelines regarding bone allografts. The following minor revisions could be made to improve the manuscript.

1.      Page 3, 3rd paragraph, “In the end, the cleansing…..osteoinductivity.”, The authors could mention some of the components in the cleansing processes, that result in reduced osteoconductivity.

2.      In the continuing sentence, the authors could briefly explain the reasons for reduced mechanical strength which leads to slower integration.

3. Section 5, the authors have discussed various literature reviews evaluating the effectiveness of mineralized bone allografts in spine fusions. They could present this in the form of a table so that readers could easily comprehend the data.

4.      The title of the manuscript should be revised to indicate that this is an opinion.

Author Response

The manuscript titled “Comparative evaluation of mineralized bone allografts for spinal surgery”, opines on the current status of the various mineralized bone allografts available for clinical use. The manuscript is easy to follow and would serve as a brief reference for understanding the FDA guidelines regarding bone allografts. The following minor revisions could be made to improve the manuscript.

  1. Page 3, 3rd paragraph, “In the end, the cleansing…..osteoinductivity.”, The authors could mention some of the components in the cleansing processes, that result in reduced osteoconductivity.

Text has been added that chemical-based agents such as hydrogen peroxide, in particular, can attenuate the osteoinductive potential of the graft.

  1. In the continuing sentence, the authors could briefly explain the reasons for reduced mechanical strength which leads to slower integration.

Text has been added noting that freeze-drying, for example, can reduce the mechanical integrity of the graft by lowering its capacity to absorb energy resulting in a more brittle graft.

  1. Section 5, the authors have discussed various literature reviews evaluating the effectiveness of mineralized bone allografts in spine fusions. They could present this in the form of a table so that readers could easily comprehend the data.

We elected not to repeat the summary of details of spine fusion trials included in the literature reviews of Kadam, Tuchman and Tavares et al as they have been previously published in detail in tabular form.  However, we did provide in Figure 2 the reported fusion rates from 8 trials separately in graphical format which have not been published previously.

  1. The title of the manuscript should be revised to indicate that this is an opinion.

The manuscript will be designated as an “Opinion” directly above the title in the upper left corner of the title page of the published manuscript.

Reviewer 2 Report

I think that discussing the topic of the use of allografts in spinal surgery is very interesting; there is not much evidence in the literature, so an article providing an overview of the topic is valuable.

The article is well written and understandable. The conclusions are consistent with the discussion of individual aspects.

However, I think one point requires more attention: the infectivological aspects. I would be grateful to the authors if they would also further discuss this topic with literature references. Thank you

Author Response

I think that discussing the topic of the use of allografts in spinal surgery is very interesting; there is not much evidence in the literature, so an article providing an overview of the topic is valuable.

We concur.

The article is well written and understandable. The conclusions are consistent with the discussion of individual aspects.

Thank you.

However, I think one point requires more attention: the infectivological aspects. I would be grateful to the authors if they would also further discuss this topic with literature references. Thank you

Text has been added to the first paragraph of Section 4 (Historical Perspective) indicating that the AATB estimated risk of allograft-associated infection is 0.014%, and that additional methods have been mandated to potentially lower the risk even further.

Reviewer 3 Report

This manuscript was an opinion article on the topic of allograft for spinal fusion.  Overall, it is a good manuscript and easy to read. However, there are some points that are needed to be considered and revised as following:

-Title: I would think it should be more specific on spinal fusion rather than spinal surgery in general.

-p.2. The storage of allograft could be preserved in many ways typically including using storage media, cryopreservation, or freeze-drying in order to store in ambient, refrigerated, or frozen states, not just only in frozen state as stated.

- Concerning the study of Tavares et al and fig 2, although the he pooled proportional rate demonstrated a high fusion rate of 87.8%. Taken this with other literature reviews, authors interpreted that allografts showed more consistency in fusion than those of DBM or CMB. However, it could be seen in Fig  2 that Lee reported a fusion rate of only 50% which was much lower than others. It would be worth commenting on this discrepancy.

-Figs 1 and 3, I think these were taken from other papers and would need to cite the references or sources along with permission from the publishers.

Author Response

This manuscript was an opinion article on the topic of allograft for spinal fusion.  Overall, it is a good manuscript and easy to read. However, there are some points that are needed to be considered and revised as following:

-Title: I would think it should be more specific on spinal fusion rather than spinal surgery in general.

The term fusion has been added to the title.

-p.2. The storage of allograft could be preserved in many ways typically including using storage media, cryopreservation, or freeze-drying in order to store in ambient, refrigerated, or frozen states, not just only in frozen state as stated.

We have added freeze-dried in addition to frozen as a typical storage method for bone allografts as specified by the AATB.

- Concerning the study of Tavares et al and fig 2, although the he pooled proportional rate demonstrated a high fusion rate of 87.8%. Taken this with other literature reviews, authors interpreted that allografts showed more consistency in fusion than those of DBM or CMB. However, it could be seen in Fig  2 that Lee reported a fusion rate of only 50% which was much lower than others. It would be worth commenting on this discrepancy.

To illustrate the associated variability in fusion rates across trials, including the 50% fusion rate reported by Lee et al, we have added the fusion rate range (50% - 100%) in the text.

-Figs 1 and 3, I think these were taken from other papers and would need to cite the references or sources along with permission from the publishers

Figure 1 is a collection of bone allograft specimens available in the public domain and obtained from the online catalogs of various tissue banks, including the musculoskeletal transplant foundation (MTF).  Figure 3 is the property of the sponsor, noted in the Conflicts of Interest statement, who supported manuscript development.